# Clinical Genetic Testing for Hearing Loss: Implications for Genetic Counseling and Gene-Based Therapies

**DOI:** 10.3390/biomedicines12071427

**Published:** 2024-06-27

**Authors:** Nam K. Lee, Kristin M. Uhler, Patricia J. Yoon, Regie Lyn P. Santos-Cortez

**Affiliations:** 1Department of Otolaryngology—Head and Neck Surgery, School of Medicine, University of Colorado Anschutz Medical Campus, Aurora, CO 80045, USA; 2Department of Physical Medicine and Rehabilitation, Children’s Hospital Colorado, Anschutz Medical Campus, University of Colorado, Aurora, CO 80045, USA; 3Department of Pediatric Otolaryngology, Children’s Hospital Colorado, Aurora, CO 80045, USA

**Keywords:** congenital, genetic panel testing, hearing loss, hereditary, novel variant, sensorineural

## Abstract

Genetic factors contribute significantly to congenital hearing loss, with non-syndromic cases being more prevalent and genetically heterogeneous. Currently, 150 genes have been associated with non-syndromic hearing loss, and their identification has improved our understanding of auditory physiology and potential therapeutic targets. Hearing loss gene panels offer comprehensive genetic testing for hereditary hearing loss, and advancements in sequencing technology have made genetic testing more accessible and affordable. Currently, genetic panel tests available at a relatively lower cost are offered to patients who face financial barriers. In this study, clinical and audiometric data were collected from six pediatric patients who underwent genetic panel testing. Known pathogenic variants in *MYO15A*, *GJB2*, and *USH2A* were most likely to be causal of hearing loss. Novel pathogenic variants in the *MYO7A* and *TECTA* genes were also identified. Variable hearing phenotypes and inheritance patterns were observed amongst individuals with different pathogenic variants. The identification of these variants contributes to the continually expanding knowledge base on genetic hearing loss and lays the groundwork for personalized treatment options in the future.

## 1. Introduction

Genetics account for 50–70% of all congenital hearing loss and the majority is non-syndromic, presenting without a set combination of clinical features in addition to hearing loss [1]. To date, over 150 genes have been identified in association with non-syndromic hearing loss, and this number continues to grow [1,2].

We continue to enhance our understanding of the complex genetic and allelic variations that influence the inheritance of hearing loss. For instance, *COL11A2* is associated with multiple inheritance patterns, including autosomal dominant, autosomal recessive, and syndromic hearing loss [3,4,5]. Additionally, non-Mendelian inheritance patterns, such as mitochondrial inheritance, modifier variants, sex-linked inheritance, and polygenic inheritance, also play significant roles [6,7,8]. This genetic heterogeneity presents clinical challenges for both diagnoses and treatment.

The genomic characterization of non-syndromic hearing loss serves as an invaluable first step for patients and families. First, understanding the disease origin can be empowering and provide comfort. Secondly, it is key to timely intervention for hearing loss. If there are any reported symptoms or conditions associated with a particular genetic variant, it provides a crucial opportunity for monitoring and treatment. The identification of causative variants further aids in the assessment of additional family members who may be at risk, as well as assistance in family planning by determining risk of hearing loss in future children.

Broadly, the association of new variants with hearing loss enhances our understanding of auditory physiology and illuminates the complication behind genetic heterogeneity in hearing loss. As our knowledge of genetic variants linked to hearing loss grows, identifying and documenting rare variants helps establish their association with hearing loss, thus aiding future diagnoses for others who carry the same variants or variants in the same gene regions. Ultimately, we can harness this information to deliver targeted personalized therapies. Multiple animal models have successfully targeted genetic underpinnings of hearing loss, and there are already clinical trials underway to target *OTOF* [9,10,11,12,13,14].

With growing advancements in sequencing technologies, genetic testing has become increasingly accessible to patients at a lower cost. Unfortunately, whole-genome sequencing remains too expensive as a clinical test for Mendelian traits, such as hearing loss. Instead, the genes identified in association with both syndromic and non-syndromic hearing loss have been harnessed in genetic panel tests for hearing loss. Gene panel testing is limited only to the genes already known to cause hearing loss and focuses on changes within the coding regions of these genes. As a result, the diagnostic yield is approximately 40%, but it serves as an invaluable first step for patients and families in understanding the disease, prognosis, and impact on their families’ overall well-being [15,16,17,18].

Here, we report putatively novel variants in the genes *MYO15A*, *MYO7A*, and *TECTA* identified through genetic panel testing in children with hearing loss at birth or in early childhood. We also discuss future trends in genetic testing for hearing loss in children.

## 2. Materials and Methods

This study was approved by the Colorado Multiple Institutional Review Board. Informed consent was obtained from parents of child participants and assent from children > 7 years old.

Patient identification occurred in partnership with the Pediatrics—Clinical Genetics and Metabolism Clinic at Children’s Hospital Colorado. Pediatric patients < 19 years old with sensorineural hearing loss (SNHL) present at birth or in early childhood and without any identifiable causes for acquired hearing loss were included. Blood or buccal swabs obtained from recruited participants were submitted for DNA isolation and the extracted DNA samples were submitted for genetic testing using a panel of hearing loss genes, i.e., 203 for Invitae (San Francisco, CA, USA) and 115 for GeneDx (Gaithersburg, MD, USA) at the time of testing (Table 1).

Patients of all ethnicities were included. In addition to demographic information (age, gender, ethnicity), past medical history such as family history of early-onset hearing loss and newborn hearing screen (NBHS) results, if available, were gathered. Clinical information on the presenting physical exam (presence of craniofacial abnormalities, ocular findings, vestibular disturbances, external ear deformities, renal abnormalities, and presence of neck mass) and radiology results were assessed.

### 2.1. Audiogram Data Review

All available audiometric data and related audiologic information were reviewed for each patient including any prior normal hearing tests, onset of hearing loss, and the degree, type, laterality, severity, and configuration of hearing loss that were documented. The clinical information on aided and unaided serial pure tone audiometry; pure tone averages (PTAs) across 500, 1000, and 2000 Hz; hearing interventions; and the speech reception threshold (SRT)/speech awareness threshold (SAT) was assessed.

The modality of hearing loss interventions, whether hearing aid fitting or cochlear implantation, was documented. When available, any tests administered to assess the patient’s speech perception abilities—pediatric AzBio (Chandler, AZ, USA), consonant–nucleus–consonant (CNC), Lexical Neighborhood Test (LNT), Bamford-Kowal-Bench Speech-in-Noise (BKB in SIN), and Early Speech Perception (ESP)—were noted to assess outcomes for respective hearing loss interventions (i.e., hearing aid, cochlear implant).

### 2.2. Genetic Testing and Variant Analysis

The genetic panel test results for hearing loss were reviewed for each patient, assessing affected genes, variants, and prediction for the pathogenicity per variant. The annotation steps taken by Invitae and GeneDx are summarized in Appendix A. A literature review was conducted for the affected genes to evaluate any associated hearing loss phenotype for the gene and/or variant.

An updated analysis of each variant’s pathogenicity was conducted independently by first converting the genome coordinate of the variants from GRCh37/hg19 to GRCh38/hg38. The converted coordinates of the variants were used to create a .vcf file, which was then annotated using ANNOVAR [19]. Then, the annotated variants were filtered for rare variants that are predicted to be pathogenic. For each variant reported by Invitae or GeneDx, the following were rechecked from updated annotations (Appendix A): (1) minor allele frequency (MAF) in any population in public genome databases including gnomAD v4.0 (hg38) or the Greater Middle East (GME) Variome [20,21,22,23,24]; (2) predicted to be deleterious by at least one bioinformatics tool included in dbSNFP42c in the case of missense or stop variants [25,26,27]; or (3) predicted to be deleterious by MutationTaster for splice and frameshift variants [28]. Ranked scores from the deleterious annotation of genetic variants using neural networks (DANN) [29] and scaled Combined Annotation Dependent Depletion (CADD) scores were also noted [10].

## 3. Results

Six pediatric patients without any identifiable cause for acquired hearing loss underwent genetic panel testing (Table 1). The average age of patients at which the genetic testing was performed was 4.5 years (standard deviation: 5.5 years). Three patients had an onset of SNHL at birth. Four out of six patients were male. Two patients identified as Hispanic, two as White, and two had unknown ethnicity. Patient 6 had testing performed by GeneDx and patients 1–5 had testing from Invitae.

### 3.1. Clinical Profiles

#### Audiologic

Two patients had a family history of congenital hearing loss in an older brother (patient 2) or a second cousin (patient 6; Table 2). Four patients did not pass their NBHS. One patient, #4, developed a vision defect in his teens (Table 1). Four patients had either MRI or CT imaging of their temporal bone, but none had any abnormal findings of the eighth nerve, internal auditory canal, or cochlea. Half of the patients presented with hearing loss at birth, one presented at age 2 years, and two presented at age 3 years. All had bilateral hearing loss.

Three patients with severe, severe-to-profound, or profound hearing loss underwent simultaneous bilateral cochlear implantation. One patient had bilateral mild-to-moderate SNHL, a second patient had bilateral moderate-to-moderately severe SNHL, and a third had moderate-to-severe SNHL on the right and mild-to-severe SNHL on the left. These latter three patients were all fitted with hearing aids.

Patient 1 was found to have congenital bilateral SNHL, not passing his newborn hearing screen and subsequent tests (Table 2). His hearing loss was severe-to-profound in both ears with a pure tone average of 92 dB on both sides (Figure 1a). His otoacoustic emissions (OAEs) were absent bilaterally, and his auditory brainstem response (ABR) and SAT were 95 and 90, respectively, on both sides. The patient had normal MRI results. This patient underwent cochlear implantation at 8 months of age at an outside institution, where he had his subsequent follow-up visits. Unfortunately, any audiometric data following the cochlear implantation could not be obtained.

Patient 2 did not pass her NBHS and was found to have a sloping moderately severe-to-profound SNHL in the right ear with a PTA of 77 and a flat severe SNHL in the left with a PTA of 83 dB (Table 2). OAEs were absent and ABR was 80 bilaterally. The patient had a family history of early-onset hearing loss in an older brother. She had a normal CT and MRI (Table 1). This patient underwent cochlear implantation at 10 months of age. Her SAT with hearing aids, and a pre-cochlear implant, was 65, which in her most recent measurement improved to 20 (right ear) and 25 (left ear) > 25 years after cochlear implantation. Her post-implant pure tone audiometry improved over time (Figure 1b).

Patient 3 passed his NBHS and presented after not passing a hearing screening at school at 3 years of age. His hearing loss was previously unsuspected but he was found to have bilateral sloping mild-to-moderate hearing loss with PTA of 27 (right ear) and 32 (left ear) (Table 2). He had absent OAEs and SRTs of 30, bilaterally. Both his pure tone audiometry and SRTs did not undergo significant changes over time (Figure 1c). This patient was fitted with hearing aids and had PBK of 90–100 (right ear) and 75–90 (left ear), and pediatric AzBio of 95.

Patient 4 did not pass his NBHS but was reported to have subsequently normal ABR. At 2 years of age, he received an audiology evaluation due to speech and language delay. He was found to have bilateral moderate-to-moderately severe SNHL (Table 2). The PTAs averaged up to date are 52 (right ear) and 50 (left ear). His OAEs at around 1 month of age were present in all frequencies, but at 28 months, they were absent bilaterally. The patient was offered *GJB2* testing, but the family deferred. At around 15 years of age, the patient developed retinitis pigmentosa, at which point genetic panel testing was performed. His hearing did not undergo significant changes over time (Figure 1d). SRT averaged 45.7 (range: 40–55) on the right and 46.4 (range: 40–50) on the left. In recent speech perception testing, pediatric AzBio was 97.8, CNC words were 88, and BKB-SIN was 2.

Patient 5 passed her NBHS but was admitted to the neonatal intensive care unit (NICU) for respiratory issues requiring intubation and her hearing was not retested at the time. She began therapy with speech–language pathology due to delay in speech and language and was found to have hearing loss at 3 years of age. She had bilateral sloping moderate-to-severe SNHL with PTAs of 52 (right ear) and 55 (left ear) on clinical presentation. She was fitted with hearing aids and has not had significant changes in her hearing over time (Figure 1e). At 4 years of age, she had ESP of 4. She had LNTs of 84 (right ear) and 80 (left ear) at 5 years.

Patient 6 did not pass his NBHS and was found to have bilateral profound SNHL with PTA of 65 (Table 2). He subsequently underwent cochlear implantation at 23 months. Post-cochlear implant PTA averaged 28 on the right and the left. His SAT also improved from 45 (bilateral) to 18 (bilateral) after the intervention.

### 3.2. Genetic Panel Testing

Variants in 18 different genes were identified in the six patients with hearing loss using genetic panel testing (Table 3). These genes include *MYO15A*, *DMXL2*, *OTOG*, *GJB2*, *ADGRV1*, *ALMS1*, *KARS*, *BTD*, *LRP2*, *MYO7A*, *PEX26*, *USH2A*, *COL4A4*, *SLC26A4*, *TMC1*, *VCAN*, *MITF*, and *TECTA*. There were 23 different variants within these genes, of which 10 were deemed pathogenic based on the combined information on variant MAF and bioinformatic prediction (Appendix A), as well as classification criteria of the American College of Medical Genetics (ACMG), with the rest being variants of uncertain significance (VUS) or benign [31].

Of the 23 variants in 18 genes, 4 were frameshift and 19 were single-nucleotide variants (SNVs). Three of the four frameshift variants are known to be pathogenic and likely causal of hearing loss in the patients. An SNV within *BTD* had a MAF > 0.01. Of the 18 SNV variants with MAF ≤ 0.01, two stop and four missense variants were considered most likely to contribute to hearing loss (Table 1 and Appendix A). Two SNVs in *MYO7A* and *TECTA* were reclassified from VUS to pathogenic (Table 1).

When selecting which variants were most likely to be causal of SNHL in each patient based on a known mode of inheritance for the gene and similarity to the reported phenotype for the patient versus in the literature, the following known variants were selected: patient 1, compound heterozygous loss-of-function variants in *MYO15A*, with one of the variants, c.9109G>T (p.Glu3037*), considered as novel; patient 2, homozygous for *GJB2* c.35delG; and patient 4, compound heterozygous loss-of-function variants within *USH2A.* A reanalysis of variants based on updated information on variant MAF and bioinformatic prediction led to the following novel autosomal dominant variants to be reclassified as pathogenic: patient 3, *MYO7A* c.2543G>A (p.Arg848Gln); and patient 6, *TECTA* c.2266 A>G (p.(Lys756Glu)). Patient 5 is heterozygous for a *TMC1* variant, c.928A>G (p.(Thr310Ala)); however, bioinformatics analyses did not have strong results towards deleteriousness. Note that patient 5 is heterozygous for a known pathogenic *SLC26A4* variant but no second variant has been identified so far. Additionally, among the six patients, only patient 5 has a strong history of potential postnatal hearing loss due to admission to the NICU.

Additional in silico analyses further support the pathogenicity of the novel variants identified in *MYO15A*, *MYO7A*, and *TECTA* (Appendix A). However, the evidence for pathogenicity of the novel *TMC1* variant remains weak.

## 4. Discussion

Technological advances in genomic sequencing expanded the affordability, and thereby the accessibility, of genetic testing to a wider audience and is rapidly expanding our understanding of variability in our genetic sequences in concert. With increased accessibility, more patients can identify the cause of their disease states. As more variants are identified in association with hearing loss, genetic testing can help families by providing an explanation of the cause, and possibilities for their children. Families will have the opportunity to prepare psychologically and also plan ahead for future follow-up and possible interventions [58,59]. The wide spectrum of benefits from an early intervention with cochlear implants in children with congenital hearing loss is well established and has been shown to impact language development, academic performance, and quality of life [60,61,62]. Furthermore, personalized therapy is not far from our future. Currently, there are three clinical trials of gene therapies targeting *OTOF* [63]. As genetic tests identify additional pathogenic variants, we are afforded with opportunities for better understanding of gene function, therefore discovering ways for targeted treatment.

Currently, the diagnostic rate of genetic panel testing rests around 40% [15,16,17,18]. Although there is an exponential growth in the number of known variants through the availability of genetic testing, our ability to interpret the significance of these variants in disease pathogenesis continues to evolve. The continued testing and subsequent expansion of genetic databases are key to improving our ability to provide genetic diagnoses for patients. This is clinically impactful in genetic counseling and development of targeted therapies through gene therapy. The patients in our cohort were able to utilize genetic panel tests to better understand the possible genetic basis for their hearing loss.

Patient 1 was found to have congenital bilateral severe-to-profound SNHL, after not passing his NBHS. Two variants, c.7124_7127del (p.Asp2375Valfs*41) and c.9109G>T (p.Glu3037*), were found in *MYO15A*, which are inherited in an autosomal recessive fashion. Pathogenic variants in *MYO15A* usually present as congenital bilateral severe-to-profound SNHL, much like our patient [32]. For the variant c.7124_7127del (p.Asp2375Valfs*41), two different hearing loss phenotypes have been reported. Both normal-to-steeply sloping-to-severe SNHL that is progressive and childhood-onset bilateral severe–profound SNHL have been observed with this variant [15,33]. The second *MYO15A* variant, c.9109G>T (p.Glu3037*), is an unreported variant. Phenotypically, the hearing loss profile of patient 1 presents most similarly to the known phenotype for *MYO15A* variants.

Patient 2 did not pass her NBHS and was found to have moderately severe-to-profound SNHL on the right and severe SNHL on the left. She has a biological older brother also with congenital hearing loss, diagnosed in Mexico, and uses sign language for communication. A well-known variant in *GJB2*, c.35delG (p.Gly12Valfs*2), was found to be homozygous in this patient, attributing to her phenotype. *GJB2* is the most commonly affected gene in non-syndromic hereditary hearing loss in many regions of the world, particularly in Europe, the Middle East, North Africa, and the countries affected by colonization, such as the Americas, and is known to result in congenital bilateral profound SNHL [37,64,65]. Her phenotype was consistent with the *GJB2* variant, which presents as moderate-to-profound bilateral SNHL [37]. Her hearing and speech improved bilaterally over time after cochlear implantation.

Patient 3 passed his NBHS but was found to have hearing loss in school testing at 3 years old. He was unsuspected of hearing loss prior to this screening and his audiogram showed bilateral mild-to-moderate SNHL. To date, he has not developed any additional symptoms. SNHL due to *MYO7A* variants, although associated with Usher Syndrome, can also present in an autosomal dominant fashion, leading to non-syndromic post-lingual moderate, progressive SNHL [66]. In these and many other cases, one cannot make final conclusions on the correlation of the clinical profile to a variant in a specific gene due to some of the clinical symptomologies present later in life, as was the case in the following patient [67].

Patient 4 did not pass his NBHS but had subsequent normal ABRs. Then, at age 2, due to his speech and language delay, his hearing was re-evaluated and he was found to have bilateral moderate-to-moderately severe SNHL. The patient and family were offered a molecular analysis for *GJB2* but decided not to proceed with the testing. However, at age 15, he developed retinitis pigmentosa when the genetic testing was performed. He was found to carry two variants in *USH2A*—c.13130C>A (p.Ser4377*) and c.2299del (p.Glu767Serfs*21). Variants in *USH2A* are associated with Usher Syndrome, type 2A, which, much like this patient, has a characteristic presentation of congenital moderate-to-severe SNHL and progressive retinitis pigmentosa that develops later in life. This patient’s hearing was monitored since the detection of his hearing loss, which had remained stable (Figure 1d). He was fitted with hearing aids at 2 years of age upon the identification of his hearing loss.

Patient 5 passed her NBHS but was admitted to the NICU after birth due to respiratory distress. She began to receive speech therapy due to speech and language delay at around 3 years of age when she was referred to audiology for a hearing test. She was found to have right-sided moderate-to-severe SNHL and mild-to-severe SNHL on the left. She had three heterozygous variants found in three different genes—*SLC25A4*, *TMC1*, and *VCAN. TMC1* variants can be inherited in both an autosomal recessive (DFNB7/11) and dominant (DFNA36) fashion in which congenital severe-to-profound SNHL and later-onset progressive hearing loss (as was in our patient) are observed, respectively [52,53]. In this case, however, we cannot entirely rule out a second *SLC26A4* variant, whether in the non-coding region or due to the copy number or structural variant(s). This patient might benefit from retesting using whole-genome sequencing, where, if still negative, they can be counseled as potentially having SNHL due to the previous NICU stay.

Patient 6 did not pass his NBHS, presenting with congenital, bilateral profound SNHL and no other symptoms or clinical findings. He was found to have two variants in two genes in his genetic panel screening for hearing loss. A heterozygous variant in *MITF*, a syndromic gene, and another variant in *TECTA* were found. *TECTA* has been identified as autosomal dominant (DFNA8/12) and autosomal recessive (DFNB21) causes of non-syndromic hereditary hearing loss. In DFNA8/12, the SNHL is mild to moderate-to-severe, and is most pronounced in the mid-frequencies [55,56]. Due to the non-syndromic nature of this patient’s SNHL in this case, *TECTA* is more likely to be causal than the *MITF* variant. Given the severity of this patient’s hearing loss, he underwent bilateral cochlear implantation with improvement in his hearing (Figure 1f).

## 5. Conclusions

There are several key points raised by the review of these six cases of genetic SNHL in children: While the initial reports from genetic testing cores are based on the discussion and consensus of in-house experts on genetic analyses and hearing loss clinicians, their decision for each patient is based on the best evidence that is available at the time of the genetic diagnosis. As variant databases covering more sequence data from additional world populations and the bioinformatic prediction of pathogenicity from new software become available, some of these diagnoses are likely to change, whether it is due to the reclassification of pathogenicity of variants, or the putatively combined effects of multiple variants on the hearing function of a single patient.Additionally, as the price of genome sequencing drops, access to the available genome data may allow for reanalyses, i.e., using updated databases and software, of other variants that lie within novel genes for SNHL, the information on which might not be available at the time of initial diagnoses. The limiting step in such a case would be reconsenting the family or patient for the reanalysis of the available genome data. There are many factors why a family might be more receptive or not to a reanalysis, though it is more likely that a previous negative report from genetic testing might favor agreement to reanalyze the genome data. Nonetheless, informed consent is essential before any reanalysis should be performed, followed by genetic counseling using the updated genetic information.Because most genetic diagnoses for SNHL are sought when a child is young, the progression of hearing loss or development of additional features (e.g., retinitis pigmentosa as part of Usher Syndrome) should trigger a request for the re-evaluation of previous genetic diagnoses [51]. While most of the literature on genetic hearing loss is focused on novel genes and variants, it will be more helpful if longitudinal analyses of hearing loss profiles and information on additional clinical features that arise over time in patients with genetic variants are published. These profiles may inform other patients with similar variants or variants within the same genes on the long-term prediction of audiologic and clinical profiles as well as the response to habilitation [68,69]. Over the past three decades, the identification of rare genetic variants for hearing loss has contributed to the cumulative information on the prevalence by population and potential mechanism of hearing and hearing loss by a gene or gene domain, which in turn facilitated the prioritization of gene therapies that are currently being developed. Updated genetic diagnoses based on the best available evidence to date will also facilitate referral once inner ear therapies for specific genes are available [70]. In the future, it will be even more important to determine pathogenicity of variants not just with bioinformatics tools but also by functional experiments to ensure that gene therapy is targeted to the pathogenic variant(s) or gene that is truly involved in the patient’s hearing loss.

## Figures and Tables

**Figure 1 biomedicines-12-01427-f001:**
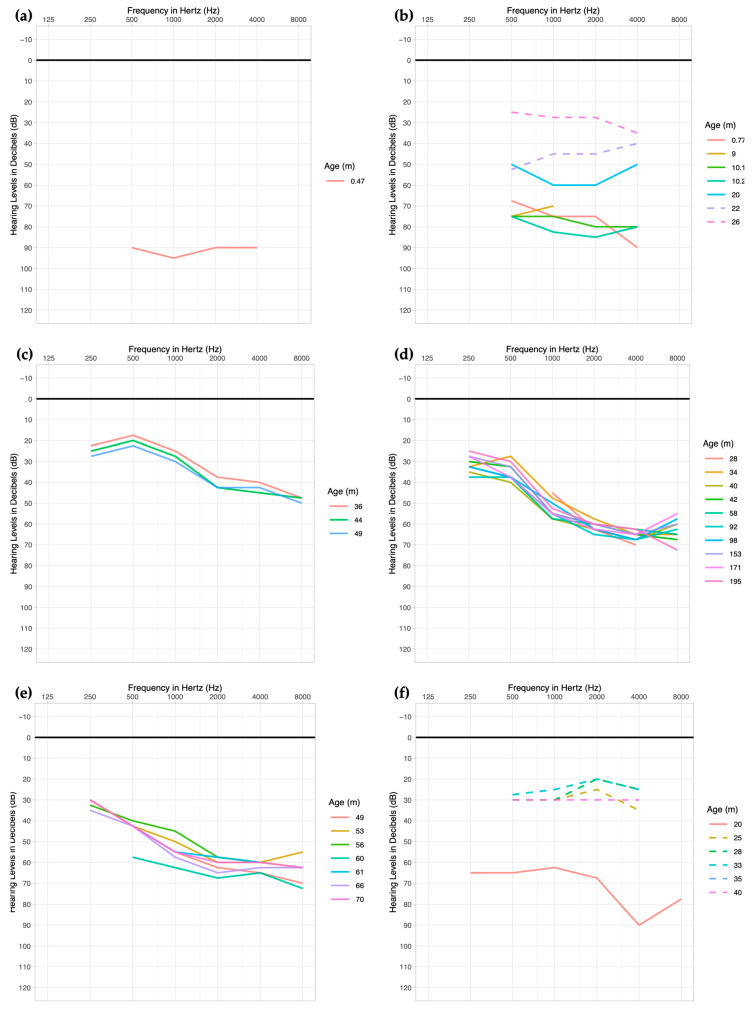
Serial pure tone average thresholds of six patients with genetic panel testing. The serial audiograms represent the average dB between the left and right ears. Audiograms for (**a**) patient 1—because this patient received care and follow-up at another facility, further audiologic information could not be obtained; (**b**) patient 2; (**c**) patient 3; (**d**) patient 4; (**e**) patient 5; (**f**) patient 6. For patients 2 and 6, *broken lines* indicate aided thresholds.

**Table 1 biomedicines-12-01427-t001:** Patients with genetic panel testing from Invitae and GeneDx.

ID	Age ^1^	Sex	Ethnicity	Other Clinical Findings ^2^	Temporal Bone Imaging	Panel
1	0.6	M	NA	None	MRI: Normal	Invitae
2	0.8	F	Hispanic	None	CT and MRI: Normal	Invitae
3	3	M	NA	None	None	Invitae
4	15	M	White	Ocular	CT: Normal	Invitae
5	6	F	White	None	None	Invitae
6	1.7	M	Hispanic	None	MRI: Normal	GeneDx

^1^ Age in years at which the patient underwent genetic testing. ^2^ Associated clinical presentations including craniofacial abnormalities, ocular findings, vestibular abnormalities, external ear malformations, and the presence of a neck mass as well as cardiac and renal findings were assessed. NA, information on ethnicity not provided by patient.

**Table 2 biomedicines-12-01427-t002:** Hearing loss profiles.

ID	Fhx	NBHS	Onset	R Severity	R Audiogram Shape	R PTA ^1^	L Severity ^2^	L Audiogram Shape	L PTA	HI	Age at HI
1	None	F	Birth	Severe-to-profound	Flat	92	Severe-to-profound	Flat	92	CI	8 mo
2	Brother	F	Birth	Moderately severe-to-profound	Sloping	77	Severe	Flat	83	CI	10 mo
3	None	P	3 yr	Mild-to-moderate	Sloping	27	Mild-to-moderate	Sloping	32	HA	3 yr
4	None	F	2 yr	Moderate-to-moderately severe	Cookie-bite	52	Moderate-to-moderately severe	Sloping	50	HA	2 yr
5	None	P	3 yr	Moderate-to-severe	Sloping	52	Mild-to-severe	Sloping	55	HA	3 yr
6	Cousin	F	Birth	Profound	Cookie-bite	65	Profound	Cookie-bite	65	CI	23 mo

^1^ Average threshold from unaided PTA was calculated from audiometric measures conducted with unaided hearing across 500, 1000, and 2000 Hz. ^2^ Normal, −10 to 15 dB; Slight, 16 to 25 dB; Mild, 26 to 40 dB; Moderate, 41 to 55 dB; Moderately severe, 56 to 70 dB; Severe, 71 to 90 dB; Profound, 91+ dB [30]. Abbreviations: Fhx, family history of early-onset hearing loss; PTA, pure tone average; HI, hearing intervention; CI, cochlear implantation; HA, hearing aid.

**Table 3 biomedicines-12-01427-t003:** Genetic panel testing results with variants, analysis of pathogenicity, and characteristic phenotype associated with identified genes or variants.

ID	Phenotype	Genes	DFN	Variant(s)	Genotype	GPT	Analysis ^a^	New Var	Gene- or Variant- Associated Phenotypes in Literature
1	Bilateral severe-to-profound SNHL present at birth	** *MYO15A* **	**DFNB3**	**c.7124_7127del (p.Asp2375Valfs*41)**	**Het**	**P**	**P**	**No**	**Gene ^b^: Congenital bilateral severe-to-profound SNHL [32]. Variant: Steeply sloping severe, progressive SNHL [33]; childhood-onset bilateral severe–profound AR SNHL [15].**
		** *MYO15A* **	**DFNB3**	**c.9109G>T (p.Glu3037*)**	**Het**	**P**	**P**	**Yes ^c^**	
		*OTOG*	DFNB18B	c.4856C>T (p.(Ser1619Leu))	Het	VUS	--	Yes	Prelingual moderate AR SNHL [34,35].
		*DMXL2*	DFNA73	c.7543A>G (p.(Met2515Val))	Het	VUS	P	Yes	Bilateral mild-to-moderate AD SNHL beginning in 20s, progressing to severe to profound in 60s [36].
2	Right moderately severe-to-profound SNHL and left severe SNHL present at birth	** *GJB2* **	**DFNB1A**	**c.35delG (p.Gly12Valfs*2)**	**Hom**	**P**	**P**	**No**	**Congenital moderate-to-profound bilateral SNHL; severity is variant-dependent [37].**
		*ALMS1*	--	c.11708G>A (p.(Arg3903Gln))	Het	VUS	--	Yes	Associated with AR Alström Syndrome; progressive bilateral moderate SNHL in childhood [38].
		*ADGRV1*	--	c.13757A>G (p.(Glu4586Gly))	Het	VUS	--	Yes	Associated with AR Usher Syndrome type IIC causing congenital moderate-to-severe SNHL [39].
		*KARS1*	DFNB89	c.1259G>A (p.(Arg420Gln))	Het	VUS	--	Yes	Bilateral, symmetric severe-to-profound or moderate-to-severe AR SNHL [40].
		*MYO15A*	DFNB3	c.9620G>A (p.Arg3207His)	Het	P	--	No	See patient 1; *MYO15A*.
3	Bilateral mild-to-moderate SNHL, onset at 3 years old	** *MYO7A* **	**DFNB2/** **DFNA11**	**c.2543G>A (p.(Arg848Gln))**	**Het**	**VUS**	**P**	**Yes**	**Severe bilateral SNHL [41,42].**
		*LRP2*	--	c.2426G>A (p.(Ser809Asn))	Het	VUS	--	Yes	Associated with AR Donnai–Barrow syndrome (Facio-oculoacousticorenal syndrome) with congenital bilateral profound SNHL though moderate SNHL was also reported [43,44]. May present as non-syndromic bilateral moderate HL in childhood [45].
		*ADGRV1*	--	c.12052G>A (p.(Val4018Ile))	Het	VUS	--	Yes	See patient 2; ADGRV1.
		*OTOG*	DFNB18B	c.7817_7820dup (p.Tyr2608Serfs*76)	Het	P	--	Yes	Prelingual bilateral moderate AR SNHL, stable throughout time [34,35].
		*PEX26*	--	c.98C>T (p.(Pro33Leu))	Het	VUS	--	Yes	Post-lingual bilateral moderate-to-severe SNHL [46]. AR Zellweger spectrum disorder results in moderately severe-to-severe SNHL [47].
		*BTD*	--	c.1270G>C (p.Asp424His)	Het	P	--	No	AR biotinidase deficiency may present with moderate-to-severe sloping SNHL within the 1st year [48]. No variant-specific hearing loss pattern reported previously.
4	Bilateral moderate-to-moderately severe SNHL, onset at 2 years old; retinitis pigmentosa at age 15	** *USH2A* **	**--**	**c.13130C>A (p.Ser4377*)**	**Het**	**P**	**P**	**No**	**Usher Syndrome, type 2—moderate to severe congenital SNHL with retinitis pigmentosa presenting at age 20–30 years [49].**
		** *USH2A* **	**--**	**c.2299del (p.Glu767Serfs*21)**	**Het**	**P**	**P**	**No**	**Variant: Variable severity of progressive HL according to variant [50].**
		*COL4A4*	--	c.980A>G (p.(Glu327Gly))	Het	VUS	--	Yes	Associated with AR Alport Syndrome with progressive mild-to-moderate bilateral SNHL that affects mid-to-high frequencies [51].
5 ^d^	Right moderate-to-severe SNHL and left mild-to-severe SNHL, onset at 3 years old	** *TMC1* **	**DFNA36/DFNB7/** **DFNB11**	**c.928A>G (p.(Thr310Ala))**	**Het**	**VUS**	**P?**	**Yes**	**Congenital severe-to-profound SNHL if AR; bilateral, symmetric SNHL that begins at 5–10 years old and rapidly progresses to profound deafness within 10–15 years if AD [52,53].**
		*SLC26A4*	DFNB4	c.1246A>C (p.Thr416Pro)	Het	P	--	No	Bilateral fluctuating or progressive moderate-to-severe AR congenital SNHL [54].
		*VCAN*	--	c.3917C>G (p.(Ala1306Gly))	Het	VUS	--	Yes	AD Wagner vitreoretinopathy.
6	Bilateral profound non-syndromic SNHL present at birth	** *TECTA* **	**DFNA8/DFNA12**	**c.2266 A>G (p.(Lys756Glu))**	**Het**	**VUS**	**P**	**Yes**	**Moderate-to-severe AD SNHL, most pronounced in the mid-frequencies [55,56].**
		*MITF*	--	c.560-7T>A	Het	VUS	P	Yes	AD Waardenburg syndrome, type II, with congenital bilateral profound SNHL [57].

^a^ GPT refers to the pathogenicity reported in the original report of the genetic panel testing, whereas Analysis refers to the pathogenicity (P) predicted in our independent analysis based on the variant MAF, deleteriousness, ACMG guidelines, and genotype given the associated AD or AR SNHL phenotype (see also Appendix A). ^b^ Gene refers to phenotypes reported in literature resulting from any variant within the gene. Variant refers to phenotypes reported in literature resulting from the specific variant the patient has. ^c^ These variants were reported to ClinVar by Invitae but not in a publication. ^d^ Patient 5 is heterozygous for a known pathogenic variant in *SLC26A4*; however, a second *SLC26A4* variant was not identified. She also stayed at the NICU. Abbreviations: DFN, non-syndromic DeaFNess loci; Het, heterozygous; Hom, homozygous; P, pathogenic; VUS, variant of uncertain significance. Among the identified variants per patient, variants in bold are the most likely to be causal of SNHL.

## Data Availability

Data is contained within the article and Appendix A. Updated annotations of the novel variants are being submitted to ClinVar.

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
