# Peer review of "Clinical Genetic Testing for Hearing Loss: Implications for Genetic Counseling and Gene-Based Therapies"

_biomedicines, 2024, doi:10.3390/biomedicines12071427_

Round 1

Reviewer 1 Report

Comments and Suggestions for Authors

Review

to manuscript “Clinical Genetic Testing for Hearing Loss: Implications for Genetic Counseling and Personalized Medicine”

by authors Nam K. Lee, Kristin M. Uhler, Patricia J. Yoon, and Regie Lyn P. Santos-Cortez

The authors presented the study about clinical and audiometric data were collected from six pediatric patients who underwent genetic panel testing.

Major comment

1. The introduction section does not contain any references to previously published works.

2. The sample size of the presented article is very small. In total only six patients were included in this original study.

3. In three patients with homozygous and compound-heterozygous P/LP variants in GJB2, MYO15A and USH2A genes causing of AR form of HL (DFNB1A, DFNB3, USH2A) it’s likely causative variants. However, in other three cases with VUS variants in genes MYO7A, TMC1 and TECTA (DFNA11, DFNA36 and DFNA8) additional evidence of their pathogenic role are required.

Recommendation

Reject

Author Response

  1. The introduction section does not contain any references to previously published works.

A reference was added to the two first sentences of the Introduction.

2. The sample size of the presented article is very small. In total only six patients were included in this original study.

This concern is understandable. However the points we wanted to underline in this report are the necessity of following up genotype-phenotype correlations in patients years after the initial genetic diagnosis to determine progression of hearing loss or habilitation outcomes, while also checking if the initial genetic diagnosis holds up over time. These follow-ups are important as gene therapies become available for various genes for hearing loss.

3. In three patients with homozygous and compound-heterozygous P/LP variants in GJB2, MYO15A and USH2A genes causing of AR form of HL (DFNB1A, DFNB3, USH2A) it’s likely causative variants. However, in other three cases with VUS variants in genes MYO7A, TMC1 and TECTA (DFNA11, DFNA36 and DFNA8) additional evidence of their pathogenic role are required.

Additional updated information regarding the three novel variants in MYO7A, TMC1 and TECTA were added to Table 1 and Supplementary Table 2. For TMC1, the genetic diagnosis is that of exclusion given that a second variant in SLC26A4 was not identified in patient 5.

Reviewer 2 Report

Comments and Suggestions for Authors

Clinical genetic testing for hearing loss:implications for genetic counseling and personalized medicine

By

Lee NK et al

The authors present six extensive case reports and the results and implications of gene panel testing (and re-analysis bioinformatically ) for hearing impairment in those six pediatric patients. The patients had undergone gene panel testing respectively by use of panel from Invitae and from GeneDx. The authors repeated the bioinformatic analysis by various bioinformatic tools and present both the classification in the initial results and in their own bioinformatic analysis. No information is given about the methodology and the bioinformatic evaluation in the results from Invitae and GeneDx is given, so it is impossible to know if the initial evaluation was too stringent /wrong or if time has elapsed and new information thereby allowed the current authors to  reclassify the variants from VUS to pathogenic.

They found compelling evidence for  pathogenic variants in MYO15A, GJB2,MYO7A, Ush 2A, TMC and TECTA in the six patients. In the case of several additional variants  found in patients 1,2,3,4, and 6, a number of variants changed classification from VUS to pathogenic, but the details of these reclassifications are not elaborated upon.

There is no information if segregation analysis (parents/sibs) have been performed to increase the conclusion of pathogenicity of the identified variants.

The paper is an extensive case report which again supports to publish variants so that over time the relevant classification becomes possible and the relevance of considering repeated GENETIC TESTING IF either none had previously been performed or if clinical features arising makes such repeated testing relevant.

A classical issue in this regard is patients with  “non-syndromic mimics” etiology of  hearing impairment (cases where the first clinical feature is apparently only hearing impairment, but where additional clinical characteristics appear later in childhood: Pendred syndrome with hypothyroidism, Usher syndrome wit retinitis pigmentosa, deafness-infertility syndrome or others.  A good reference in this regard is: Gooch C et al Int j Pediatr otorhinol150 (2021)110872.

Author Response

No information is given about the methodology and the bioinformatic evaluation in the results from Invitae and GeneDx is given, so it is impossible to know if the initial evaluation was too stringent /wrong or if time has elapsed and new information thereby allowed the current authors to  reclassify the variants from VUS to pathogenic.

They found compelling evidence for  pathogenic variants in MYO15A, GJB2,MYO7A, Ush 2A, TMC and TECTA in the six patients. In the case of several additional variants  found in patients 1,2,3,4, and 6, a number of variants changed classification from VUS to pathogenic, but the details of these reclassifications are not elaborated upon.

Additional information regarding the reclassified variants within MYO7A, TMC1 and TECTA were added to Table 1 and Supplementary Table 2. In some cases the likely causal variant is based on exclusion of other reported variants (e.g. heterozygous variant only in an autosomal recessive gene).

There is no information if segregation analysis (parents/sibs) have been performed to increase the conclusion of pathogenicity of the identified variants.

Unfortunately we did not have parental genotype information to determine compound heterozygosity of MYO15A and USH2A variants. On the other hand, the minor allele frequencies of these rare variants indicate that they are unlikely to be in linkage disequilibrium and inherited in cis. This was noted in the footnote to Supplementary Table 2.

The paper is an extensive case report which again supports to publish variants so that over time the relevant classification becomes possible and the relevance of considering repeated GENETIC TESTING IF either none had previously been performed or if clinical features arising makes such repeated testing relevant.

A classical issue in this regard is patients with  “non-syndromic mimics” etiology of  hearing impairment (cases where the first clinical feature is apparently only hearing impairment, but where additional clinical characteristics appear later in childhood: Pendred syndrome with hypothyroidism, Usher syndrome wit retinitis pigmentosa, deafness-infertility syndrome or others.  A good reference in this regard is: Gooch C et al Int j Pediatr otorhinol150 (2021)110872.

This reference was added.

Reviewer 3 Report

Comments and Suggestions for Authors

This study analyzed 6 patients with hearing loss disorders using whole-genome sequencing in addition to the general gene panel of genetic hearing loss, and found several variants in genes related to known hearing loss genes. The contents of this study may be valuable for researchers engaging in hearing loss disorders, however there are some concerns before the publication.
1. Is there any discussion on the pair of variants of the same gene? Some patients have heterozygous mutation with different locations within the same gene.
2. The title does not represent the contents of this article, so it should be chnaged.

Cases of congenital deafness in which mutations were found by whole genome analysis but not by gene panel testing

3. Gene therapy is still a work in progress, and the discussion should properly address the implications of vigorously pursuing gene testing despite the absence of treatment.

Author Response

  1. Is there any discussion on the pair of variants of the same gene? Some patients have heterozygous mutation with different locations within the same gene.

Unfortunately we did not have parental genotype information to determine compound heterozygosity of MYO15A and USH2A variants. On the other hand, the minor allele frequencies of these rare variants indicate that they are unlikely to be in linkage disequilibrium and inherited in cis. This was noted in the footnote to Supplementary Table 2.

2. The title does not represent the contents of this article, so it should be chnaged.

The title was modified as suggested.

3. Gene therapy is still a work in progress, and the discussion should properly address the implications of vigorously pursuing gene testing despite the absence of treatment.

This is a quandary that has plagued the hearing loss genetics community for decades. However given the rarity of variants involved in hearing loss, reporting these variants as they are found has helped accumulate information on gene-based prevalence of variants by population and the potential disease mechanisms by gene or gene domain, both of which contributed to prioritization of gene therapies that are currently being developed.  This was added as third to last sentence of the last paragraph of the Discussion.

Reviewer 4 Report

Comments and Suggestions for Authors

- Expand on the techniques section by giving more details on the particular genetic panel tests that were used, the number of genes they covered, the sequencing technology they utilized, and any other studies (such copy number variation analysis) that were carried out.

-Discuss several hearing loss confounders as middle ear disease. cite doi:10.3390/jcm11237000

- Provide a flowchart or diagram that shows the steps in the genetic testing and analysis workflow as well as the patient selection procedure.

- For simpler understanding, include more comprehensive patient demographics and clinical data in a structured table style. This data should include specific ethnicities, family history details, and related clinical findings.

-Talk about the possible ramifications of the newly discovered pathogenic variants in the study, such as the necessity for functional research to verify pathogenicity or the expansion of the phenotypic spectrum linked to the genes.

- Give additional background on the findings' possible effects on genetic counseling procedures and individualized techniques to treating hearing loss, like the application of gene therapy or targeted medication therapies.  Discuss and cite doi:10.3390/biomedicines11061616.

- Add a section discussing the difficulties and restrictions associated with using genetic panels to screen for hearing loss. Some of these include the need to periodically re-analyze the data in light of new information, interpret variants of unknown significance, and possible incidental discoveries.

Author Response

- Expand on the techniques section by giving more details on the particular genetic panel tests that were used, the number of genes they covered, the sequencing technology they utilized, and any other studies (such copy number variation analysis) that were carried out.

Please see Supplementary Table 1. The number of genes tested at the time the patient-specific reports were released were 203 for Invitae and 115 for GeneDx at the time of testing (last sentence, paragraph 2 of Materials and Methods). There are more genes being tested currently for either core.

-Discuss several hearing loss confounders as middle ear disease. cite doi:10.3390/jcm11237000

None of the reported cases had otitis media in their medical history. There were also no documentation of environmental risk factors for congenital hearing loss such as CMV infection or perinatal insults, except for NICU admission in patient 5.

- Provide a flowchart or diagram that shows the steps in the genetic testing and analysis workflow as well as the patient selection procedure.

Instead of adding a flowchart, we made it clearer which variants were reclassified as pathogenic or likely causal of hearing loss per patient by adding extensive variant details to Table 1 and Supplementary Table 2.

- For simpler understanding, include more comprehensive patient demographics and clinical data in a structured table style. This data should include specific ethnicities, family history details, and related clinical findings.

Please see Table 1 and the first paragraph under 3.1 Clinical Profiles.

-Talk about the possible ramifications of the newly discovered pathogenic variants in the study, such as the necessity for functional research to verify pathogenicity or the expansion of the phenotypic spectrum linked to the genes.

The following was added as last sentence: "In the future, it will be even more important to determine pathogenicity of variants not just with bioinformatics tools but also by functional experiments to ensure that gene therapy is targeted to the pathogenic variant(s) or gene that is truly involved in the patient’s hearing loss."

These two sentences were also previously included in the same paragraph: "While most literature on genetic hearing loss is focused on novel genes and variants, it will be more helpful if longitudinal analysis of hearing loss profiles and information on additional clinical features that arise over time in patients with genetic variants are published. These profiles may inform other patients with similar variants or variants within the same genes on long-term prediction of audiologic and clinical profiles as well as response to habilitation."

- Give additional background on the findings' possible effects on genetic counseling procedures and individualized techniques to treating hearing loss, like the application of gene therapy or targeted medication therapies.  Discuss and cite doi:10.3390/biomedicines11061616.

Paragraph 1 of the Discussion was expanded and this reference was also included.

- Add a section discussing the difficulties and restrictions associated with using genetic panels to screen for hearing loss. Some of these include the need to periodically re-analyze the data in light of new information, interpret variants of unknown significance, and possible incidental discoveries.

Please see paragraph 2 of the Discussion.

Round 2

Reviewer 1 Report

Comments and Suggestions for Authors

Review-round 2

to manuscript “Clinical Genetic Testing for Hearing Loss: Implications for Genetic Counseling and Personalized Medicine”

by authors Nam K. Lee, Kristin M. Uhler, Patricia J. Yoon, and Regie Lyn P. Santos-Cortez

The authors presented the study about clinical and audiometric data were collected from six pediatric patients who underwent genetic panel testing.

Major comments

1. The Introduction section is very short, formulated from common phrases and consists only one cited reference. In general, even the update Introduction does not allow evaluation of the relevance of the presented work.

2. In my opinion, the sample size (6 patients with P/LP or VUS variants in the different, previously well studied genes MYO15A, GJB2, MYO7A, USH2A, TMC1 and TECTA) do not meet for the concept of the genotype-phenotype correlation studies.

3. For the VUS variants additional evidence of their pathogenic role is required (segregation analysis in affected families, in-silico and/or functional evidence). Since the causative role of the presented VUS variants remains in doubt, I think genotype-phenotype reports of the patients with these VUS-variants are premature.

Recommendation

Reject

Author Response

  1. The Introduction section is very short, formulated from common phrases and consists only one cited reference. In general, even the update Introduction does not allow evaluation of the relevance of the presented work.

The Introduction was expanded as suggested.

  1. In my opinion, the sample size (6 patients with P/LP or VUS variants in the different, previously well studied genes MYO15A, GJB2, MYO7A, USH2A, TMC1 and TECTA) do not meet for the concept of the genotype-phenotype correlation studies.

Unfortunately, longitudinal audiometric data in a large majority of the literature on genetic hearing loss is severely lacking. Our main concepts are to go beyond cross-sectional audiometric data for genetic hearing loss or reporting of novel variants in databases and to start reporting genotype-phenotype correlations of longitudinal audiometric data on a per variant basis, with updates on pathogenicity using the newest information available. We believe this is very important for the field of genetic hearing loss going forward.

  1. For the VUS variants additional evidence of their pathogenic role is required (segregation analysis in affected families, in-silicoand/or functional evidence). Since the causative role of the presented VUS variants remains in doubt, I think genotype-phenotype reports of the patients with these VUS-variants are premature.

Supplementary Table 3 which has additional information from molecular modeling and domain analyses was added for the four novel variants. Please note that for many studies which report hundreds of variants, these types of information e.g. longitudinal audiologic data, re-analysis of pathogenicity of variants, tend to be buried or obscure because of emphasis on gene-based prevalence per population or cross-sectional analyses.

Round 3

Reviewer 1 Report

Comments and Suggestions for Authors

I am so sorry, in my opinion, the revised version of the manuscript do not meen for original genotype-phenotype studies.

Author Response

We appreciate your insights and your dedication to ensuring the quality of the research. We understand your concern of our study not meeting the standards for original genotype-phenotype studies. We have revised the wording on our abstract and parts of our discussion to avoid drawing genotype-phenotype conclusions. As you can see in our discussion, we describe the clinical context of each patient and discuss the typical phenotype presentation of the variants found in the patients, while taking care not to draw conclusion. We additionally note in our conclusion, the importance of for functional experiments in determining true correlation of the variant to the phenotype.